# Manganese Sulfanyl Porphyrazine–MWCNT Nanohybrid Electrode Material as a Catalyst for H_2_O_2_ and Glucose Biosensors

**DOI:** 10.3390/s24196257

**Published:** 2024-09-27

**Authors:** Michal Falkowski, Amanda Leda, Mina Hassani, Michal Wicinski, Dariusz T. Mlynarczyk, Nejat Düzgüneş, Michal P. Marszall, Grzegorz Milczarek, Jaroslaw Piskorz, Tomasz Rębiś

**Affiliations:** 1Department of Medicinal Chemistry, Collegium Medicum in Bydgoszcz, Faculty of Pharmacy, Nicolaus Copernicus University in Torun, Dr. A. Jurasza 2, 85-089 Bydgoszcz, Poland; 503341@doktorant.umk.pl (M.H.); mmars@cm.umk.pl (M.P.M.); 2Faculty of Health Sciences, Collegium Medicum, The Mazovian University in Płock, 09-402 Płock, Poland; 3Institute of Chemistry and Technical Electrochemistry, Poznan University of Technology, Berdychowo 4, 60-965 Poznan, Poland; amanda.leda@doctorate.put.poznan.pl (A.L.); grzegorz.milczarek@put.poznan.pl (G.M.); 4Department of Pharmacology and Therapy, Collegium Medicum in Bydgoszcz, Faculty of Medicine, Nicolaus Copernicus University in Torun, Curie Sklodowskiej 9, 85-094 Bydgoszcz, Poland; michal.wicinski@cm.umk.pl; 5Chair and Department of Chemical Technology of Drugs, Poznan University of Medical Sciences, Rokietnicka 3, 60-806 Poznan, Poland; 6Department of Biomedical Sciences, Arthur A. Dugoni School of Dentistry, University of the Pacific, San Francisco, CA 94103, USA; nduzgunes@pacific.edu; 7Chair and Department of Inorganic and Analytical Chemistry, Poznan University of Medical Sciences, Rokietnicka 3, 60-806 Poznan, Poland; piskorzj@ump.edu.pl

**Keywords:** carbon nanotube, phthalimide, manganese, porphyrazine, voltammetry, hydrogen peroxide

## Abstract

The demetallation reaction of sulfanyl magnesium(II) porphyrazine with N-ethylphthalimide substituents, followed by remetallation with manganese(II) salts, yields the corresponding manganese(III) derivative (Pz3) with high efficiency. This novel manganese(III) sulfanyl porphyrazine was characterized by HPLC and analyzed using UV-Vis, MS, and FT-IR spectroscopy. Electrochemical experiments of Pz3 conducted in dichloromethane revealed electrochemical activity of the new complex due to both manganese and N-ethylphthalimide substituents redox transitions. Subsequently, Pz3 was deposited on multiwalled carbon nanotubes (MWCNTs), and this hybrid material was then applied to glassy carbon electrodes (GC). The resulting hybrid electroactive electrode material, combining manganese(III) porphyrazine with MWCNTs, showed a significant decrease in overpotential of H_2_O_2_ oxidation compared to bare GC or GC electrodes modified with only carbon nanotubes (GC/MWCNTs). This improvement, attributed to the electrocatalytic performance of Mn^3+^, enabled linear response and sensitive detection of H_2_O_2_ at neutral pH. Furthermore, a glucose oxidase (GOx)-containing biosensing platform was developed by modifying the prepared GC/MWCNT/Pz3 electrode for the electrochemical detection of glucose. The bioelectrode incorporating the newly designed Pz3 exhibited good activity in the presence of glucose, confirming effective electronic communication between the Pz3, GOx and MWCNT surface. The linear range for glucose detection was 0.2–3.7 mM.

## 1. Introduction

The rapid development of society increases the burden of better understanding of human health. Unfortunately, not every aspect of prophylaxis and treatment can be fully explored, as there is often a lack of analytical devices or techniques that could provide rapid, qualitative, and reliable data, which can also be produced cheaply and in a sustainable manner. Because of this, new sensing devices are sought to overcome such shortcomings. One of such promising approaches is to utilize electrochemical sensors [1].

Enzymatic electrochemical glucose biosensors are the most popular devices for glucose monitoring available on the market. The research on the application of enzymes towards glucose sensing have been carried out widely over the past decades. Glucose biosensors monitor the redox current generated when electrons are transferred either directly or indirectly between an enzymatic receptor and a conducting electrode surface [2].

Generally, three electrochemical concepts have been applied to determine glucose levels by using enzymes immobilized on the electrode surface. The first approach involves enzyme-catalyzed reactions of glucose in the presence of oxygen in which glucose is oxidized to gluconic acid. Such a process primarily comprises the formation of electroactive hydrogen peroxide. The glucose concentration can be determined by electrochemically measuring the H_2_O_2_ released due to the enzymatic reaction. The second approach is based on using a reversible, diffusible, or immobilized redox mediator that allows the transfer of electrons from the GOx to the electrode. The concept of the third-generation glucose biosensors utilizes a reagentless approach, and the direct electron transfer between the enzyme and the electrode generates the current (i.e., analytical signal) [3,4].

However, there is still plenty of room for improvement in the development of enzyme electrochemical biosensors. The devices described above mostly suffer from insufficient reusability, narrow linear range, and low storage stability. All of these issues must be addressed to increase their commercial applicability and efficiency of use. For instance, the need for a highly-efficient hydrogen peroxide catalyst is still very high. Porphyrinoids, due to their interesting and often unique properties, including high catalytic rate, are often the components of such new devices [5]. The naturally occurring porphyrin ring and its synthetic aza-analogues, porphyrazine (Pz) and phthalocyanine (Pc), are macrocyclic structures offering a highly conjugated π-bond system. Additionally, the four pyrrole rings forming the macrocycle enable the complexation of cations, including transition metal cations, which can actively contribute to the electrochemical properties of the molecules [6,7,8]. Additionally, it was established that, among the mentioned macrocyclic compounds, Pzs offer the best electrocatalytic properties [9]. The unsubstituted porphyrazine ring, however, is a planar aromatic moiety, which tends to aggregate and is poorly soluble. Therefore, the derivatization of the Pz ring is necessary [10]. Porphyrazines with sulfanyl substituents were found not only to exhibit favorable optical and electrochemical properties, but also to have potential in photodynamic therapy of cancer and microbes, as well as in the selective detection of hydrogen peroxide and other biologically-relevant molecules [11,12,13,14]. Furthermore, sulfanyl Pzs bearing phthalimide substituents that have been explored so far, also by our group, have the potential for being components of electrochemical sensors [15,16,17]. The strongly non-polar chemical structure of Pzs may also be considered beneficial; due to their aromatic structure, porphyrinoids can be attached on the surface of carbon nanostructures, including single-walled carbon nanotubes (SWCNTs), MWCNTs, or graphene layers via simple and efficient noncovalent adsorption. Such an elegant approach enables the fabrication of hybrid nanomaterials with high electrocatalytic properties towards many analytes, such as nitrite, hydrazine, hydrogen peroxide, or ascorbic acid [18,19].

Based on the above considerations, this study aimed to explore the transition metal complex of sulfanyl porphyrazine bearing phthalimide substituents that would be capable of hydrogen peroxide detection. The Mn^3+^ complex was synthesized by means of chemical modification of the metal-free derivative and characterized using mass spectrometry, UV-Vis spectrophotometry, and infrared spectroscopy. The newly synthesized Pz was subjected to electrochemical studies and was deposited on MWCNTs. As a result, we obtained a promising electrode material revealing high electrocatalytic ability towards hydrogen peroxide oxidation. Moreover, the proposed hybrid nanomaterial was considered as a platform for immobilization of glucose oxidase. The resultant biosensor material was evaluated for glucose determination.

## 2. Materials and Methods

### 2.1. General Procedures

The glassware used for the chemical reactions was oven-dried and the reactions were performed in an inert gas atmosphere. Rotary-evaporation was performed at or below 60 °C under reduced pressure. The reported reaction temperatures refer to Radleys^®^ Heat-On (Essex, UK) display values. Flash column chromatography was performed using Merck silica gel, 40–63 μm particle size (Darmstadt, Germany) with eluents (dichloromethane and methanol) provided by POCh (Gliwice, Poland). Thin-layer chromatography (TLC) used silica gel Merck Kieselgel 60 F254 (Darmstadt, Germany) plates. Mass spectra (MALDI-TOF) were recorded at the European Center for Bioinformatics and Genomics in Poznan. For the UV-Vis spectrophotometry experiments a Shimadzu UV-1900i (Shimadzu, Kyoto, Japan) spectrophotometer was used, as well as high-precision 1 cm light path quartz cells (Hellma Analytics). Fourier-transform infrared spectra were recorded on the IRAffinity-1 spectrometer (Shimadzu, Kyoto, Japan), in the range 500–4000 cm^−1^ using KBr as a blank. All the used chemicals were reagent grade and no additional purification was required. An Agilent 5500 AFM microscope (Santa Clara, United States) and Hitachi S-3400N Scanning Electron Microscopes (Tokyo, Japan) were used to examine the surface morphology of MWCNT and MWCNT/**Pz3** materials. 

### 2.2. Synthetic Procedures and Characterization

**2,3,7,8,12,13,17,18-Octakis[(*N*-ethylphthalimide)thio]porphyrazinato magnesium(II) (Pz1)** and **2,3,7,8,12,13,17,18-octakis[(*N*-ethylphthalimide)thio]porphyrazine** (**Pz2**) were synthesized using a previously published procedure [16].

**2,3,7,8,12,13,17,18-Octakis[(*N*-ethylphthalimide)thio]porphyrazinato manganese(III) chloride** (**Pz3**).

Porphyrazine **2** (45 mg, 0.023 mmol), MnCl_2_ × 4H_2_O (45 mg, 0.23 mmol) and DMF (8 mL) were loaded into a round-bottom flask and the reaction mixture was stirred at 75 °C for 24 h. After cooling to room temperature, the reaction mixture was evaporated to dryness. The deep violet residue was dissolved in DCM (8 mL), after which 8 mL of saturated NaCl aqueous solution was added and the resulting two-phase mixture was stirred at room temperature for 45 min exposed to fresh air. The aqueous fraction was discarded and the organic layer was dried with MgSO_4_ and rotary evaporated. The dark violet residue was chromatographed (dichloromethane/methanol, gradient 100:1 to 20:1, *v*/*v*) to give a dark violet film of porphyrazine **3** (40 mg, yield = 87%). R*_f_* (dichloromethane/methanol, 100:1, *v*/*v*) 0.30. UV-Vis (dichloromethane) λ_max_ nm (log ε) 290 (4.62), 335 (4.46), 516 (4.39), 717 (4.50). MS (MALDI) *m*/*z*: 2007.2 [M]^+^. FT-IR (KBr, *ν*_max_/cm^−1^): 3198s, 2961w, 2930m, 2855m, 2378w, 2317w, 1771s (C=C), 1714s (C=N, C=O, amide), 1467m (aromatics), 1433m, 1394s (aromatics), 1361m, 1324m, 1295m, 1260w, 1188m, 1171w, 1085s, 1042s (maleimides), 1008w, 976m, 932m, 865m, 797s (aromatic C-H), 717s (aromatics), 610w, 531m. HPLC purity ~100.0% (see Appendix A).

### 2.3. Reagents and Materials for Electrochemical Analysis

Multi-walled carbon nanotubes (MWCNT, an average diameter: 10 nm; an average length: 1.5 μm) were supplied by Metrohm DropSens (Oviedo, Spain). Merck (Darmstadt, Germany) delivered the tetrabutylammonium perchlorate (TBAP), dichloromethane (DCM, puriss. p.a., ≥99.9%), *N*,*N*-dimethylformamide (anhydrous 99.8%), hydrogen peroxide (H_2_O_2_) stock solution (30 wt. %), and Glucose Oxidase from *Aspergillus niger* (GOx, ≥100,000 units g^−1^, 160 kDa). D(+)-glucose monohydrate (Glu, ≤99.5%), produced by Carl Roth (Karlsruhe, Germany), was used to prepare the glucose solution. Fructose, lactose, maltose, and saccharose (for research as interferents) were purchased from Warchem (Warszawa, Poland). POCH (Gliwice, Poland) provided the monosodium (NaH_2_PO_4_) and disodium (Na_2_HPO_4_) phosphates necessary to prepare the phosphate buffer (PB, pH 7.4). Potassium hexacyanoferrate(III) (K_3_[Fe(CN)_6_]) was supplied also by POCH (Gliwice, Poland). Ferrocene (Fc) was provided by Alfa Aesar (Haverhill, MA, USA). Since each of these compounds was of reagent quality, further purification was not required.

### 2.4. Electrochemical Measurments

An EmStat4S electrochemical analyzer (Eindhoven, The Netherlands) was used for all electrochemical experiments. Electrochemical impedance spectroscopy (EIS) measurements were conducted at an applied potential of 0.2 V, with a duration of 20 s. The frequency range for the tests spanned from 0.1 Hz to 100,000 Hz, and a potential amplitude of 0.1 V was used. Ag/AgCl (3 M KCl) was utilized as the reference electrode in aqueous electrolyte and platinum wire was used as the counter electrode in the three-electrode setup. A glassy carbon electrode (GC, diameter: 3 mm) was chosen as the working electrode (BASi), West Lafayette, IN, USA. Silver wire with deposited AgCl (Ag/AgCl) was employed as the reference electrode for organic experiments conducted in 0.1 M TBAP in DCM. Following each experiment, ferrocene (Fc) was introduced to determine the electrode potentials in relation to the ferrocenium/ferrocene pair (Fc^+^/Fc).

### 2.5. Fabrication of GC/MWCNT, GC/MWCNT/Pz3, and GC/MWCNT/Pz3/GOx Modified Electrodes

The GC was polished on a polishing cloth with an aqueous suspension of aluminum oxide (Al_2_O_3_, Buehler, Lake Bluff, IL, USA average diameter: 50 nm) before starting each electrochemical measurement. Any impurities were then eliminated using an ultrasonic bath which included an acetone/water solution (1:1, *v*/*v*). After this, the electrode was cleaned and the surface was drop-cast with 10 μL of MWCNT dispersion (1 mg mL^−1^ in DMF). Next, the electrode was oven-dried at 60 °C until the solvent evaporated. The GC/MWCNT electrode was submerged into the **Pz3** solution in DCM (1 mg mL^−1^). By soaking the electrode in **Pz3**, the porphyrazine was immobilized noncovalently. An enzyme stock solution was first prepared at 1 mg mL^−1^ in 0.05 mol L^−1^ phosphate buffer (pH 7.4) and stored at −18 °C. To prepare the GC/MWCNT/**Pz3**/GOx modified electrode, the enzyme was thawed at 4 °C, and then the electrode was immersed in the GOx solution for 30 min under stirring conditions. After the enzyme adsorbed on the electrode surface, it was washed with distilled water [20,21]. 

To perform electrochemical testing, every electrode was positioned in the intended electrolyte. Before experimentation, a glass cell with a supporting electrolyte was deoxygenated using N_2_ gas. During the experiments with glucose solutions, the system was oxygenated. Every electrochemical measurement was conducted at room temperature (approx. 25 °C).

## 3. Results and Discussion

### 3.1. Synthesis and Characterization

The synthetic route leading to the manganese(III) porphyrazine bearing phthalimide moieties was developed. First, the magnesium(II)-containing derivative (**1**) and its free-base analog (**2**) were synthesized [16]. **Pz2** was reacted with manganese(II) chloride tetrahydrate which led to manganese(III) sulfanyl porphyrazine **3** formation (Figure 1). All intermediates and the final product were purified using flash column chromatography. The purity of **Pz3** was determined based on the HPLC analyses performed in three different solvent systems which confirmed the purity of the isolated molecule (see Appendix A). Following this, **Pz3** was characterized by mass spectrometry, as well as FT-IR and UV-Vis spectroscopy.

### 3.2. Optical Properties

The electronic absorption spectra of porphyrazine **3** were recorded in three organic solvents: dichloromethane (DCM), DMF, and dimethylsulfoxide (DMSO) (Figure 1). The manganese(III) cations in the porphyrazine core highly influenced the intensity and position of the absorption bands.

Manganese(III) porphyrazine **3** revealed a Soret band of moderate intensity, with differences in position and profile, depending on the solvent (λ_Abs_ in the range 355–383 nm). The Q band of **Pz3** was low-intensive, broad, and in the case of DCM and DMSO divided into two sub-bands with λ_Abs_ of 516 and 717 nm for DCM and 588 and 712 nm for DMSO. In DMF, only one Q-band maximum at 668 nm was observed. The logarithms of the molar absorption coefficients of **Pz3** Q-bands were between 4.35 and 4.50) (Figure 1, Table 1).

### 3.3. Electrochemical Study of Pz3 in Organic Electrolyte

The newly synthesized **Pz3** was assessed for its electrochemical properties by means of the cyclic voltammetry (CV) and differential pulse voltammetry (DPV). The investigation was conducted in dichloromethane as the solvent with the addition of tetrabutylammonium perchlorate (TBAP, 0.1 M). The results are summarized in Figure 2. **Pz3** exhibited four well-defined redox peaks in organic electrolyte. In our previous studies we investigated the electrochemical response of iron(II) and cobalt(II) porphyrazines bearing phthalimide groups [17]. We found that the peaks III and IV are correlated with the redox transitions of the macrocyclic ring (peak III) and phthalimide substituents (IV). Additionally, in Figure 2A,B we observe the reversible peaks at −0.49 V (peak I) and −0.98 V (peak II). The existence of two metal-based redox couples were observed previously for macrocyclic compounds bearing manganese ions [22,23]. Such peaks can be ascribed to manganese electroactivity originating from Mn(III)/Mn(II) (peak I) and Mn(II)/Mn(I) (peak II) transitions.

### 3.4. SEM and AFM Study of MWCNT and MWCNT/Pz3 Material

Figure 2 illustrates the procedure used to construct the modified electrode for biosensing applications, which involves the deposition of MWCNT, adsorption of **Pz3**, and immobilization of GOx. The morphology of the electrode materials was analyzed using SEM and AFM techniques. The SEM images in Figure 3a,b show that the MWCNTs exhibit a random distribution, forming a porous network. A comparison of the SEM images revealed no significant changes in the morphology of the MWCNTs following modification with **Pz3**. Both MWCNT and MWCNT/**Pz3** retained their fibrous tubular structure, without noticeable aggregation or concentration points.

The topography of MWCNT and MWCNT/Pz3 was further analyzed using AFM at varying magnifications (Figure 4). To assess the changes in MWCNT dimensions following **Pz3** deposition, we measured the diameters of the nanotubes shown in Figure 4a,b. The average diameter of MWCNT/Pz3 is slightly larger than that of the unmodified nanotubes, measuring 24.5 ± 3.7 nm (n = 10), compared to 22.7 ± 5.21 nm (n = 10) for pristine MWCNT. These observations confirm a strong interaction between **Pz3** and MWCNT via π–π stacking, resulting in a uniform and smooth layer on the MWCNT surface.

### 3.5. Electrochemical Characterization of Pz3 Deposited on MWCNT (GC/MWCNT/Pz3) in Phosphate Buffer (PB)

To examine the electrochemical activity, the synthesized **Pz3** was immobilized on the surface of MWCNT. The voltammetric responses of the designed modified electrodes are presented in Figure 5.

The GC/MWCNT electrode (curve a, marked by a black line) shows a typical capacitive character without redox peaks. In the case of the GC/MWCNT/**Pz3** modified electrode, peaks corresponding to the electrochemical transformations of phthalimide substituents can be distinguished in the range of negative potentials: the anode peak at a potential of −0.4 V and the cathode peak at −0.5 V. Electrochemical transformations of phthalimide substituents at the mentioned potentials in PB were also observed in our previous studies [17,24]. Moreover, in Figure 5, a reversible Mn^3+^/Mn^2+^ transition can be distinguished at a formal potential of approximately 0.05 V. The Mn^3+^/Mn^2+^ transition at a similar potential, recorded in PB (pH = 7.4) was observed previously by Agboola and Nyokong [23]. Following the integration of the anodic peak associated with the transition from Mn^2+^ to Mn^3+^ and taking into account the peak charge, the estimated metal loading was determined to be 46.48 ng. It is plausible to hypothesize that the aromatic structure of phthalimide facilitates additional π–π interactions between the porphyrazine and the MWCNT. This interaction may lead to a firmer attachment of **Pz3** onto the modified electrode’s surface, thereby enhancing electron transfer efficiency. Hence, the strong interactions between **Pz3** and MWCNT may be favorable in practical applications of the material in the development of electrocatalytic sensors. To assess the electrocatalytic effect towards the oxidation of hydrogen peroxide, voltammograms were obtained in the presence of 2 mM H_2_O_2_. H_2_O_2_ is a co-product of enzymatic reactions; hence much attention of scientists is focused on the search for catalysts for the oxidation of this analyte [25]. The performance of the constructed GC/MWCNT/**Pz3** electrode was evaluated in comparison to bare GC and GC/MWCNT electrodes. In the case of bare GC, hydrogen peroxide reduction required a significantly negative overpotential, resulting in a relatively small observed reductive current (Appendix A). When utilizing a GC/MWCNT electrode, a slight enhancement in H_2_O_2_ redox behavior was noted, with both cathodic and anodic current waves observed (Appendix A). However, the voltammetric curve indicated electrochemical irreversibility. Furthermore, the GC/MWCNT/**Pz3** modified electrode exhibits notable electrocatalytic efficacy, as illustrated in Figure 6 (curve b), where the addition of 2 mM H_2_O_2_ leads to a significant enhancement of the redox peaks. Both reduction and oxidation of hydrogen peroxide occur at a low overpotential and relatively high current on the GC/MWCNT/**Pz3** electrode. These findings indicate that the GC/MWCNT/**Pz3** electrode is promising in the context of electrocatalysis of this analyte.

CV measurements were conducted across various scan rates ranging from 10 to 100 mV s^−1^ in phosphate buffer (PB, pH 7.4) to evaluate the electron transport kinetics on the surface of the GC/MWCNT/**Pz3** modified electrode (Figure 7A). In Figure 7B, I vs. v relationships are plotted for all anodic and cathodic peaks (for phthalimide substituents and manganese oxidation/reduction). The peak currents exhibited a linear increase with the scan rate, suggesting adsorption of **Pz3** on the MWCNT surface. The π–π stacking interaction between the conjugated porphyrazine macrocycle and the extensively delocalized π-bonding network of the carbon nanomaterial enable the surface-limited redox characterization.

Electrochemical impedance spectroscopy (EIS) measurements were conducted on the electrode surface to gain deeper insights into interfacial processes. Figure 8A displays representative Nyquist plots for bare GC, GC/MWCNT, and GC/MWCNT/**Pz3** in the presence of 1 mM [Fe(CN)_6_]^3−/4−^ (1:1 molar ratio). The EIS data are correlated with the cyclic voltammetry (CV) results, which are shown as an inset in the figure. The Nyquist plot for bare GC (curve a) exhibits a relatively large semicircle in the high-frequency region, indicative of significant charge transfer resistance, followed by a straight line in the low-frequency region, representing diffusion resistance. In contrast, the EIS curves for GC/MWCNT (curve b) and GC/MWCNT/**Pz3** (curve c) show predominantly straight lines in the low-frequency region, signifying rapid electron transfer and diffusion-controlled processes on the electrode surfaces. The absence of a semicircle in the high-frequency region for GC/MWCNT and GC/MWCNT/**Pz3**, compared to the bare GC electrode, indicates enhanced electron transfer between [Fe(CN)_6_]^3−/4−^ and the electrode material. The deposition of Pz on the MWCNT surface does not impede electron transfer. Figure 8B illustrates also the comparative cyclic voltammetry for three types of electrodes: unmodified GC electrode, GC/MWCNT, and GC/MWCNT/**Pz3**, in the presence of 1 mM K_3_[Fe(CN)_6_)]. Modification with MWCNT led to a notable increase in peak currents compared to the bare GC electrode, attributed to the well-known highly porous structure and good electron transfer properties of MWCNT [26,27]. For the bare GC electrode, the cathodic peak current was −4.43 µA, while after MWCNT modification it was −12.14 µA. The addition of manganese ion-metalized porphyrazine to the GC/MWCNT system further augmented peak currents (−13.02 µA). This enhancement is likely due to the strong π–π interactions between the synthesized porphyrazine and the MWCNT surface, which slightly increases the porosity of the MWCNT structure.

Table 2 summarizes the peak-to-peak separation values for the studied electrodes. These separations support the previously mentioned observations, demonstrating slow electron transfer kinetics for the bare GC electrode and a notable reduction in peak separation for the modified electrodes (GC/MWCNT and GC/MWCNT/Pz3). The bare GC electrode exhibited a peak-to-peak separation of 94 mV, which exceeds the theoretical value of 60 mV for a one-electron reversible diffusion-controlled process. However, such high peak separations are commonly observed for bare GC electrodes [17,23]. In contrast, the peak-to-peak separations for the GC/MWCNT and GC/MWCNT/Pz3 electrodes decreased to 67 and 68 mV, respectively, indicating enhanced electron transfer kinetics at the MWCNT-modified surface. The surface coverage with the **Pz3** does not significantly affect the electron transfer kinetics, which is favorable for the application of GC/MWCNT/**Pz3** in electrochemical sensors.

The cyclic voltammetry results at various scan rates, conducted in phosphate buffer (pH 7.4) containing 1 mM K_3_[Fe(CN)_6_], along with the corresponding calibration curves for each electrode modification, are presented in Appendix A. These results facilitated the determination of the electroactive surface area of the electrodes. The electroactive surface areas were calculated for the mentioned electrodes (Table 2). The Randles–Sevcik equation was used for this purpose [28]:I_p_ = 2.69∙10^5^∙A∙D^1/2^∙n^3/2^∙C∙v^1/2^
(1)
where I_p_ denotes the peak current [A]; A stands for the electroactive surface area [cm^2^]; D is the diffusion coefficient, which is equal to 7.3∙10^−6^ cm^2^ s^−1^ for the ferricyanide; n denotes the transferred electron number, which in this case is equal 1; v denotes the scan rate [V s^−1^]; and C is the molar concentration [mol cm^−3^]. Covering the GC electrode with MWCNT results in more than four times increase of the electroactive surface. This effect is stronger after the next modification with **Pz3**. For the GC/MWCNT/**Pz3** modified electrode, the electrode surface is the largest (0.208 cm^2^), which indicates that such an electrode has the greatest potential as an electrocatalyst.

### 3.6. Electrochemical Measurements of the GC/MWCNT/Pz3 Electrode in the Presence of Hydrogen Peroxide

Figure 9 displays CVs obtained at a scan rate of 10 mV s^−1^ for GC/MWCNT/**Pz3** with increasing concentrations of H_2_O_2_. Clear anodic waves emerge and grow with successive H_2_O_2_ additions. Anode currents rise linearly from 78.7 µM to 18.2 mM for the manganese-ion-modified porphyrazine electrode, proving the electrode’s capability for H_2_O_2_ measurement across a wide concentration range.

To monitor hydrogen peroxide, the chronoamperometric measurements were applied. The measurement was carried out at the potential E_1_ = +0.1 V for 10 s, and E_2_ = +0.3 V for 60 s (Figure 10A). As expected, at the E_1_ potential the electrode shows no response to the addition of H_2_O_2_, while at the E_2_ potential a linear increase in current is noticeable after the addition of hydrogen peroxide (from 38.5 to 6081 µM). The detection of hydrogen peroxide on the presented electrode allowed for the determination of basic analytical parameters. It was calculated that the limit of detection (LOD) is 10.4 µM, the limit of quantification (LOQ) is 31.6 µM, and the sensitivity of such an electrode is 0.01496 µA µM^−A^ cm^−c^ (14.96 µA mM^−1^ cm^−2^).

The chronoamperometric studies regarding the GC/MWCNT/**Pz3** electrode towards H_2_O_2_ was performed in a PB solution (pH 7.4) under constant stirring. The working potential of +0.3 V was applied. After the addition of each subsequent portion of hydrogen peroxide to PB, current–time curves were recorded (Figure 11A), leading to a rapid increase of the oxidation current, eventually stabilizing at a constant level. These findings suggest that GC/MWCNT/**Pz3** possesses robust electrocatalytic properties, facilitating the measurement of hydrogen peroxide oxidation at low potentials. Calibration curves were generated by plotting the amperometric response against the concentration of H_2_O_2_ (Figure 11B). For the present system, linearity is observed over the entire tested range, i.e., from 9.9 µM to 4930 µM. The estimated LOD for this technique was 2.9 µM and LOQ was 8.7 µM. The sensitivity of such an electrode was calculated and is equal to 0.0524 µA µM^−1^ cm^−2^ (52.4 µA mM^−1^ cm^−2^).

Table 3 lists the analytical parameters of electrodes designated GC/MWCNT/**Pz3** for the detection of hydrogen peroxide, together with others previously reported in the literature. The parameters were obtained using chronoamperometric techniques (under stagnant and stirring conditions). Many parameters have comparable values when compared to similar works. The main advantage of the modified electrode constructed in this work is the relatively wide range of linearity, allowing the determination of H_2_O_2_ at low concentrations (of the order of µM) as well as at higher concentrations (mM).

### 3.7. Glucose Biosensing at GC/MWCNT/Pz3/GOx

To verify the applicability of GC/MWCNT/**Pz3** as an electrocatalyst in glucose biosensors, we immobilized GOx on the surface and we tested the electroanalytical performance for glucose detection. First, we decided to compare the chronoamperometric response of glucose addition for the GC/MWCNT/**Pz3**/GOx-modified electrode and the porphyrazine-free electrode (GC/MWCNT/GOx). As can be seen in Appendix A, the lack of **Pz3** leads to no response of the electrode towards glucose oxidation. This control experiment allows us to conclude that the glucose biosensor built based on the modified GC/MWCNT/**Pz3**/GOx electrode will have good operating parameters.

Figure 12 displays double-step chronoamperometric responses recorded at the GC/MWCNT/**Pz3**/GOx electrode under addition of increased glucose aliquots. A linear increase in current is observed after the addition of glucose, ranging from 0.2 to 3.7 mM. The enzymatic reaction between the GOx active center (FAD) and glucose, as well as subsequent oxidation of released H_2_O_2_ at GC/MWCNT/**Pz3**/GOx, can be represented by Equations (2)–(4):Glucose + GOx(FAD^+^) → Glucolactone + GOx(FADH_2_) (2)

In the presence of oxygen, FAD^+^ is regenerated, leading to the formation of hydrogen peroxide:GOx(FADH_2_) + O_2_ → GOx(FAD^+^) + H_2_O_2 _
(3)

Hydrogen peroxide is oxidized by the manganese porphyrazine catalyst (GC/MWCNT/**Pz3**/GOx):H_2_O_2_ → 2H^+^ + O_2_ + 2e^−^
(4)

The course of the curve in the entire tested concentration range also resembles the kinetics of the Michaelis–Menten reaction, which is characteristic of enzymatic systems (Figure 12B) [34]. The detection of enzymatically produced hydrogen peroxide on the GC/MWCNT/**Pz3**/GOx electrode allowed the determination of basic analytical parameters. It was calculated that the LOD is 0.05 mM, LOQ is 0.15 mM, and the sensitivity of such an electrode is 2.73 µA mM^−1^ cm^−2^ (for the determined area of the GC/MWCNT/**Pz3**/GOx electrode equal to A = 0.2063 cm^2^, according to the equation described above).

The chronoamperometric response of the GC/MWCNT/**Pz3**/GOx electrode was examined also concerning glucose at stirring conditions in PB solution (pH 7.4) at an operation potential of +0.6 V. Figure 13A illustrates typical current curves over time for subsequent additions of glucose. Upon introducing glucose into the PB, the oxidation current swiftly escalates, eventually reaching a stable level. These results indicate that GC/MWCNT/**Pz3**/GOx possesses electrocatalytic properties conducive to measuring glucose oxidation at low concentrations. Calibration curves were constructed by plotting the amperometric response against the glucose concentration. Within the tested range of 0.47 mM to 3.24 mM, linearity was observed. The system is characterized by a typical enzymatic calibration curve (Michaelis–Menten reaction kinetics) [34,35]. The estimated LOD and quantification LOQ for this technique were 0.14 mM and 0.41 mM, respectively. The sensitivity of such an electrode was calculated to be 1.29 µA mM^−1^ cm^−2^ (for the determined area of the GC/MWCNT/**Pz3**/GOx electrode equal to A = 0.2063 cm^2^, according to the equation described above). For the curve shown in Figure 13B, a linear approximation of a Lineweaver–Burk plot may be applied. The Michaelis–Menten constant (K_m_) was calculated to be 9.27 mM. A low K_m_ value indicates a high affinity of the enzyme for its substrate and is comparable to the value reported in [36].

Table 4 shows a comparison of the performance of the obtained biosensor for glucose determination with others available in the literature. The parameters obtained here have comparable values in relation to the cited works. As the greatest advantage of our biosensor, we would like to highlight the linearity range covering relatively low glucose concentrations (thus enabling the determination of this analyte in samples with a low sugar content, e.g., food samples or human blood).

Appendix A displays the study of reproducibility of GC/MWCNT/**Pz3**/GOx electrodes. Cyclic voltammograms of the seven newly prepared electrodes were recorded in PB buffer at 50 mV s^−1^. The calculated RSD is equal 4.6% (n = 7) and suggest satisfactory electrode-to-electrode reproducibility.

The response of GC/MWCNT/**Pz3**/GOx to a 1 mM solution of glucose was monitored over the course of 2 h (Appendix A). The glucose addition resulted in a sharp but temporary spike in current to 4.20 μA, which decreased with time. The percentage decrease in signal after 1 h was calculated at 21% based on these data. After 2 h, this value was equal about 37%. The amperometric response was monitored in the presence of potentially interfering substances to determine the selectivity of the GC/MWCNTs/**Pz3**/GOx electrode. Specifically, other sugars commonly encountered in the food industry, such as fructose, lactose, maltose, and saccharose, were chosen for this evaluation, each added at a concentration of 1 mM. The outcomes of this assessment are illustrated in Appendix A. Although the addition of 1 mM glucose prompted a rapid current response, none of the introduced interfering substances elicited a current response. Hence, the acquired biosensor demonstrates satisfactory selectivity under the specified conditions.

In the last stage of the research, the system was used to analyze real samples of selected energy drinks containing sugars (including glucose) that was purchased from the local supermarket. The tests were performed by adding a standard to the diluted drink sample. The standard addition curve is presented in Appendix A. The standard addition was a 1 mM glucose solution. Three replicates were performed (RSD = 0.7%). The glucose content was determined at 7.49 g/100 mL. The level of all sugars declared by the manufacturer is 11 g/100 mL.

## 4. Conclusions

In the course of this study, a novel manganese(III) porphyrazine peripherally substituted with (*N*-ethylphthalimide)thio moieties was synthesized. Upon purification, it was subjected to physicochemical characterization using mass spectrometry, UV-Vis spectrophotometry and IR spectroscopy. The obtained porphyrazine showed characteristic Soret and Q-bands in the absorption spectra. The prepared macrocycle was subjected to electrochemical studies and was deposited on MWCNT, thus providing a new complex nanomaterial. We have shown in this work that the GC/MWCNT electrode modified by manganese(III)-bearing porphyrazine exhibits valuable electrocatalytic activity towards the oxidation of hydrogen peroxide. After immobilization of GOx, the application of GC/MWCNTs/**Pz3**/GOx for the amperometric determination of glucose was evaluated. A linear relationship between the current and glucose concentration was observed for the presented biosensor, with a sensitivity of 2.73 µA mM^−1^ cm^−2^ and a linear range between 0.2 and 3.7 mM. According to the data, the novel **Pz3** is a compound of choice for the development of electrochemical sensors of hydrogen peroxide or glucose.

## Data Availability

Data is contained within the article and Appendix A.

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
