# Peer review of "Manganese Sulfanyl Porphyrazine–MWCNT Nanohybrid Electrode Material as a Catalyst for H2O2 and Glucose Biosensors"

_sensors, 2024, doi:10.3390/s24196257_

Round 1

Reviewer 1 Report

Comments and Suggestions for Authors

The paper investigates the potential use of a hybrid electroactive electrode material, combining manganese(III) porphyrazine with MWCNTs, to create a sensor to quantitatively detect glucose. The authors undertook all phases of the research, from the synthesis of the materials of interest to the fabrication of the electrode modified with MWCNTs decorated with manganese(III) porphyrazine. An extensive investigation was conducted using cyclic voltammetry, HPLC, UV-VIS, and FT-IR. The results demonstrated the glucose detection capability within a linear range of 0.2 to 3.7 mM.

The article is well-structured, with each section presented in a clear and concise manner, effectively conveying relevant results, and providing adequate discussion. Considering the significance and potential applications of the research presented in this manuscript, as well as its overall quality, I express my support for its publication in the Journal of Sensors. 

However, I recommend the incorporation of a study using SEM and, if possible, an EDS map for better visualization of the morphology of carbon nanotubes with adhered porphyrazine, as it is common to observe the formation of large agglomerates of MWCNT-based materials.

Author Response

The paper investigates the potential use of a hybrid electroactive electrode material, combining manganese(III) porphyrazine with MWCNTs, to create a sensor to quantitatively detect glucose. The authors undertook all phases of the research, from the synthesis of the materials of interest to the fabrication of the electrode modified with MWCNTs decorated with manganese(III) porphyrazine. An extensive investigation was conducted using cyclic voltammetry, HPLC, UV-VIS, and FT-IR. The results demonstrated the glucose detection capability within a linear range of 0.2 to 3.7 mM. 

We sincerely appreciate the Reviewer’s valuable comments and feedback. We trust that the revisions made will meet the Reviewer’s expectations. 

1.The article is well-structured, with each section presented in a clear and concise manner, effectively conveying relevant results, and providing adequate discussion. Considering the significance and potential applications of the research presented in this manuscript, as well as its overall quality, I express my support for its publication in the Journal of Sensors.  

  1. However, I recommend the incorporation of a study using SEM and, if possible, an EDS map for better visualization of the morphology of carbon nanotubes with adhered porphyrazine, as it is common to observe the formation of large agglomerates of MWCNT-based materials.

The morphology of the electrode materials was characterized using SEM and AFM techniques. SEM images in Fig. 3a and 3b (new in the manuscript) reveal that the MWCNTs are randomly distributed, forming a porous network. A comparison of the SEM images shows no significant changes in MWCNT morphology after modification with Pz3, with both MWCNT and MWCNT/Pz3 retaining their fibrous tubular structure without noticeable aggregation or concentration points. Further analysis of the topography of MWCNT and MWCNT/Pz3 was conducted using AFM at various magnifications (Fig. 4). To evaluate any changes in MWCNT dimensions after Pz3 deposition, the diameters of the nanotubes in Fig. 4a and Fig. 4b were measured. The average diameter of MWCNT/Pz3 was found to be slightly larger than that of the unmodified nanotubes, at 24.5 ± 3.7 nm (n = 10) compared to 22.7 ± 5.21 nm (n = 10) for pristine MWCNT. These results suggest a strong interaction between Pz3 and MWCNT via π-π stacking, resulting in a uniform and smooth coating on the MWCNT surface. Unfortunately, we were unable to perform EDS analysis due to a lack of access to the necessary equipment. 

Reviewer 2 Report

Comments and Suggestions for Authors

This paper shows Manganese sulfanyl porphyrazine–MWCNT nanohybrid electrode material as a catalyst for H2O2 and glucose biosensing, I think a major revision is needed. The details are as follows.

1.    It is suggested to add an experimental flow chart for readers to understand.

2.    It is suggested that the author indicate the meaning represented by each curve in the experimental diagram of the article, similar to Figure 1 in the article, to facilitate readers' understanding.

3.    The article mentions that Fig. 5 proves that Pz3 is adsorbed on the surface of MWCNT. It is recommended to supplement the characterization of the morphology before and after adsorption, and to observe and further prove that Pz3 is adsorbed.

4.    Comparing the i shown by b and c in Figure 6, it can not be directly seen that the i shown by c is larger than b. It is suggested to give the value of i and supplement the EIS experiment for further demonstration.

5.    The electroactive surface areas of different modified electrodes are listed in Table 2, but the relevant experimental diagrams are not shown. It is recommended to supplement the support materials.

6.    Table 4 shows the different parameters of different biosensors for glucose detection, and it is recommended to refer to some articles in the past 3 to 5 years for comparison.

Comments on the Quality of English Language

Minor editing of English language required.

Author Response

This paper shows Manganese sulfanyl porphyrazine–MWCNT nanohybrid electrode material as a catalyst for H2O2 and glucose biosensing, I think a major revision is needed. The details are as follows. 

We sincerely appreciate the Reviewer’s valuable comments and feedback. We trust that the revisions made will meet the Reviewer’s expectations. 

  1. It is suggested to add an experimental flow chart for readers to understand. 

The experimental flow chart was added. Please see Scheme 2. 

  1. It is suggested that the author indicate the meaning represented by each curve in the experimental diagram of the article, similar to Figure 1 in the article, to facilitate readers' understanding. 

Thank you very much for your comment. Each of the figures presented in the manuscript and the supplement has symbols explaining their meaning. These are either legends placed directly on the graph or curve captions placed in the graph caption. 

  1. The article mentions that Fig. 5 proves that Pz3 is adsorbed on the surface of MWCNT. It is recommended to supplement the characterization of the morphology before and after adsorption, and to observe and further prove that Pz3 is adsorbed. 

The morphology of the electrode materials was analyzed using SEM and AFM techniques. The SEM images in Fig. 3a and 3b (new in manuscript) show that the MWCNTs exhibit a random distribution, forming a porous network. A comparison of the SEM images revealed no significant changes in the morphology of the MWCNTs following modification with Pz3. Both MWCNT and MWCNT/Pz3 retained their fibrous tubular structure, without noticeable aggregation or concentration points.  The topography of MWCNT and MWCNT/Pz3 was further analyzed using AFM at varying magnifications (Fig. 4). To assess the changes in MWCNT dimensions following Pz3 deposition, we measured the diameters of the nanotubes shown in Fig. 4a and Fig. 4b. The average diameter of MWCNT/Pz3 is slightly larger than that of the unmodified nanotubes, measuring 24.5 ± 3.7 nm (n = 10), compared to 22.7 ± 5.21 nm (n = 10) for pristine MWCNT. These observations confirm a strong interaction between Pz3 and MWCNT via π-π stacking, resulting in a uniform and smooth layer on the MWCNT surface. 

  1. Comparing the i shown by b and c in Figure 6, it can not be directly seen that the i shown by c is larger than b. It is suggested to give the value of i and supplement the EIS experiment for further demonstration. 

During the voltammetric studies conducted in the presence of a redox marker (1 mM K₃[Fe(CN)₆]), we compared the reduction (cathodic) currents, as the oxidized form of the marker was used. The CV scan was initiated at a potential of 0.6 V and progressed towards less positive values. The cathodic peak current observed for the GC/MWCNT/Pz electrode was slightly higher than that of the GC/MWCNT electrode, suggesting a modest increase in the electroactive surface area, likely due to an increase in the porosity of the GC/MWCNT/Pz electrode. The manuscript has been updated to include the cathodic current values, along with the results of the EIS experiments, which are also provided. 

  1. The electroactive surface areas of different modified electrodes are listed in Table 2, but the relevant experimental diagrams are not shown. It is recommended to supplement the support materials. 

The supplement includes the appropriate CV plots and calibration curves for each electrode modification. Please see the new Fig. S2. 

  1. Table 4 shows the different parameters of different biosensors for glucose detection, and it is recommended to refer to some articles in the past 3 to 5 years for comparison. 

Articles from several previous years were also added to the table and compared. 

Round 2

Reviewer 2 Report

Comments and Suggestions for Authors

Accept.